# Temporal fluctuations of correlators in integrable and chaotic quantum systems

Talía L. M. Lezama,[1] Yevgeny Bar Lev,[2] and Lea F. Santos[3]

[1]*Department of Physics, Yeshiva University, New York, New York 10016, USA*
[2]*Department of Physics, Ben-Gurion University of the Negev, Beer-Sheva 84105, Israel*
[3]*Department of Physics, University of Connecticut, Storrs, Connecticut 06269, USA*

We provide bounds on temporal fluctuations around the infinite-time average of out-of-time-ordered and time-ordered correlators of many-body quantum systems without energy gap degeneracies. For physical initial states, our bounds predict the exponential decay of the temporal fluctuations as a function of the system size. We numerically verify this prediction for chaotic and interacting integrable spin-1/2 chains, which satisfy the assumption of our bounds. On the other hand, we show analytically and numerically that for the XX model, which is a noninteracting system with gap degeneracies, the temporal fluctuations decay polynomially with system size for operators that are local in the fermion representation and decrease exponentially in the system size for non-local operators. Our results demonstrate that the decay of the temporal fluctuations of correlators cannot be used as a reliable metric of chaos or lack thereof.

## I. INTRODUCTION

The spreading of local observables under unitary time evolution has been used as a measure of information scrambling in quantum systems out-of-equilibrium and has spurred a lot of interest across various areas of physics. A central quantity used to assess this spreading is the out-of-time ordered commutator (OTOC) between two operators, $\hat{W}(t)$ and $\hat{V}(0)$, where one is fixed at time $t = 0$ and the other evolves in time, that is,

$$
\begin{aligned}
C(t) &= -\frac{1}{2}\langle[\hat{W}(t), \hat{V}]^2\rangle \\
&= 1 - \Re[\langle\hat{W}^\dagger(t)V^\dagger(0)W(t)V(0)\rangle],
\end{aligned}
\tag{1}
$$

where $\Re[.]$ indicates the real part of the out-of-time-ordered correlation function $\langle\hat{W}^\dagger(t)V^\dagger(0)W(t)V(0)\rangle$. Initially, the commutator is small or zero, and the spreading of the operator in time is manifested in the growth of the commutator.

The OTOC was introduced more than half a century ago in the semiclassical analysis of superconductivity [1], and recently attracted a lot of attention in high energy physics [2–9], random unitary circuits [10–13], diffusive dynamics [14–17], many-body localization [18–21], quantum phase transitions [22–25], integrable models [26], quantum chaos [27–48], and instability [49–54]. The interest in this quantity has inspired a number of experimental studies [55–67].

For chaotic systems with a well-defined semiclassical limit, the initial growth of the OTOC with time is exponential, with a rate determined by the positive Lyapunov exponents of the corresponding classical system [1]. As such, OTOCs are natural candidates to explore the chaotic features of a given system. However, this initial exponential growth happens also in integrable models due to instability [49–54], as experimentally confirmed in [67]. There have also been examples of interacting-integrable systems without a well-defined semiclassical limit, where the OTOC exhibits a diffusive front broadening as in nonintegrable models [16, 17], and

of chaotic models with local conserved quantities, where the OTOC grows algebraically [11, 15, 68–70]. Recently, it was also shown that for a class of many-body local circuits, exponential OTOC decay is not a good indicator of chaos [71].

While the short-time behavior of the OTOC does not categorically distinguish between integrable and chaotic quantum systems, one may wonder whether its long-time behavior could. In [36], the authors used the size of the temporal fluctuations after the saturation of the OTOC as a way to differentiate between chaos and integrability.

Motivated by [36] and various other studies on the temporal fluctuations of observables [72–79] and OTOCs [40, 41], we investigate how the magnitude of the temporal fluctuations of time-ordered correlation functions and out-of-time-ordered correlation functions depend on the system size $L$. We obtain analytical bounds that show that for systems without energy gap degeneracies (chaotic or not), the fluctuations decay at least exponentially with the system size. We confirm this result numerically by considering three spin-1/2 models in one-dimensional (1D) lattices: a chaotic model with first- and second-neighboring couplings, the integrable interacting XXZ model, and the integrable noninteracting XX model. In the first two cases, energy gap degeneracies are absent, so the scaling of the fluctuations with $L$ cannot set them apart and the fluctuations decay exponentially with $L$. For the XX model, where energy gap degeneracies are present, the decay can be polynomial or exponential depending on the local operators used in the correlators.

The paper is structured as follows. In Sec. II, we introduce the correlators that we study and the general bounds for the decay of their temporal fluctuations as a function of system size. In Sec. III, we present the models and initial states that we consider. In Sec. IV, we study numerically the decay of the fluctuations with $L$ and verify that our bounds are tight. In Sec. V, we present analytical results for the XX model, for which the general bounds do not apply. In Sec. VI, we summarize our main results and outline possible research directions.

Derivations and supporting material are provided in the appendices.

## II.   CORRELATORS AND BOUNDS ON THEIR TEMPORAL FLUCTUATIONS

We consider the time-ordered correlation function,

$$F_2^A(t) = \langle \Psi_0 | \hat{A}(t) \hat{A} | \Psi_0 \rangle, \tag{2}$$

and the out-of-time-ordered correlation function,

$$F_4^A(t) = \langle \Psi_0 | \hat{A}(t) \hat{A} \, \hat{A}(t) \hat{A} | \Psi_0 \rangle, \tag{3}$$

where $|\Psi_0\rangle$ is an initial state, $\hat{A}$ is a local operator, which in our case is Hermitian and unitary, $\hat{A}(t) = e^{i\hat{H}t} \hat{A} e^{-i\hat{H}t}$, and $\hat{H}$ is the Hamiltonian of the system.

We investigate the infinite-time average of the correlators,

$$\overline{F_{2,4}^A} = \lim_{T \to \infty} \frac{1}{T} \int_0^T dt \, F_{2,4}^A(t), \tag{4}$$

and the magnitude of their temporal fluctuations around this value,

$$\Delta_{F_{2,4}^A}^2 \equiv \overline{\left| F_{2,4}^A(t) - \overline{F_{2,4}^A} \right|^2}. \tag{5}$$

### A.   Bounds on Fluctuations

To obtain general bounds on the temporal fluctuations of $F_{2,4}^A$, we generalize the results of Refs. [74–76] for the fluctuations in an observable $\langle \hat{A}(t) \rangle$. Our results for $F_2^A$ are complementary to those in Ref. [41], where a bound on the fluctuations of $F_2^A$ was obtained for thermal initial states and for systems exhibiting weak ETH, as well as to those in Refs. [80, 81], where the fluctuations of k-time-ordered correlation functions were bounded on average (see also [82] for studies on temporal fluctuations of nonequilibrium currents).

The Hamiltonian associated with the evolution of the correlators has eigenvalues $E_n$ and eigenstates $|E_n\rangle$. It is a Hermitian operator that can be written as $\hat{H} = \sum_n E_n \hat{P}_n$, where $\hat{P}_n = \sum_{q=1}^{K_n} |E_{n_q}\rangle\langle E_{n_q}|$ is a projector onto the degenerate subspace with $K_n$ equal eigenvalues $E_n$, and all $E_n$'s in the sum for $\hat{H}$ are distinct by construction.

#### 1.   Fluctuations of the time-ordered correlation function

Using the projectors, the time-ordered correlation function in Eq. (2) becomes

$$F_2^A(t) = \sum_{n,m} e^{i(E_n - E_m)t} \langle \Psi_0 | \hat{P}_n \hat{A} \hat{P}_m \hat{A} | \Psi_0 \rangle. \tag{6}$$

Since all $E_n$'s are distinct, the infinite-time average is

$$\overline{F_2^A} = \sum_n \langle \Psi_0 | \hat{P}_n \hat{A} \hat{P}_n \hat{A} | \Psi_0 \rangle, \tag{7}$$

and the fluctuations around $\overline{F_2^A}$ are given by

$$\Delta_{F_2^A}^2 = \sum_{n \neq m} \sum_{k \neq l} e^{i(E_n - E_m)t} e^{-i(E_k - E_l)t} \tag{8}$$

$$\times \langle \Psi_0 | \hat{P}_n \hat{A} \hat{P}_m \hat{A} | \Psi_0 \rangle \langle \Psi_0 | \hat{A}^\dagger \hat{P}_l \hat{A}^\dagger \hat{P}_k | \Psi_0 \rangle. \tag{9}$$

We now assume that the energy gaps are non-degenerate, which means that if the gaps $E_n - E_m = E_k - E_l$ for any given $n, m, k, l$, then either $n = m$ and $k = l$ or $n = k$ and $k = l$. Since $n = m$ and $k = l$ are excluded from the sums in Eq. (9), we obtain the following expression for the infinite-time averaged fluctuations in Eq. (5),

$$\begin{aligned}
\Delta_{F_2^A}^2 &= \sum_{n \neq m} \langle \Psi_0 | \hat{P}_n \hat{A} \hat{P}_m \hat{A} | \Psi_0 \rangle \langle \Psi_0 | \hat{A}^\dagger \hat{P}_m \hat{A}^\dagger \hat{P}_n | \Psi_0 \rangle \\
&\leq \sum_{n,m} \langle \Psi_0 | \hat{P}_n \hat{A} \hat{P}_m \hat{A} | \Psi_0 \rangle \langle \Psi_0 | \hat{A}^\dagger \hat{P}_m \hat{A}^\dagger \hat{P}_n | \Psi_0 \rangle \\
&= \mathrm{tr}\left( \hat{A} \omega_A \hat{A}^\dagger \omega \right),
\end{aligned} \tag{10}$$

where we defined,

$$\omega_A = \sum_n \hat{P}_n \hat{A} | \Psi_0 \rangle \langle \Psi_0 | \hat{A}^\dagger \hat{P}_n \tag{11}$$

$$\omega = \sum_n \hat{P}_n | \Psi_0 \rangle \langle \Psi_0 | \hat{P}_n. \tag{12}$$

Using the Cauchy-Schwarz inequality, Eq.(10) gets bounded as (see Appendix A),

$$\Delta_{F_2^A}^2 \leq \left\| \hat{A} \right\|^4 \sqrt{\mathrm{tr}(\omega^2)}, \tag{13}$$

where $\|A\|$ is the matrix norm corresponding to the largest eigenvalue of $\hat{A}$. The quantity $\mathrm{tr}(\omega^2)$ corresponds to the inverse participation ratio (IPR) of the initial state in the basis of the eigenstates of the Hamiltonian,

$$\mathrm{tr}(\omega^2) = \mathrm{IPR}_0 = \sum_{n_q, q} |C_{n_q}^{(0)}|^4, \tag{14}$$

where $C_{n_q}^{(0)} = \langle E_{n_q} | \Psi_0 \rangle$.

The result in Eq. (13) implies that for Hamiltonians with non-degenerate energy gaps and initial conditions that have weight on exponentially many eigenstates of the Hamiltonian (which may happen when the Hamiltonian describes a many-body quantum system), the fluctuations decay exponentially with the system size. It is important to stress that, similarly to Refs. [74, 76], the obtained bound does not require the system to be chaotic or the spectrum to be non-degenerate.

### 2. Fluctuations of the out-of-time-ordered correlation function

To bound the fluctuations of $F_4^A(t)$, we first introduce the following simplified notation,

$$T_{nmkl} \equiv \left\langle \Psi_0 \left| \hat{P}_n \hat{A} \hat{P}_m \hat{A} \hat{P}_k \hat{A} \hat{P}_l \hat{A} \right| \Psi_0 \right\rangle, \qquad (15)$$

$$S_{nmkl} \equiv E_n - E_m + E_k - E_l.$$

The infinite-time average of $F_4^A(t)$ is given by

$$\overline{F_4^A} = \sum\nolimits'_{nmkl} T_{nmkl}, \qquad (16)$$

and the fluctuations around the infinite-time average are

$$\Delta_{F_4^A}^2 = \sum\nolimits'_{n,m,k,l} \sum\nolimits'_{n',m',k',l'} T_{nmkl} T_{n'm'k'l'}^* \\ \times \delta \left( S_{nmkl} - S_{n'm'k'l'} \right), \qquad (17)$$

where $\delta(x)$ is a Kronecker delta and the prime indicates that all terms with $S_{nmkl}, S_{n'm'k'l'} = 0$ are not included in the sums. The Kronecker delta implies that $S_{nmkl} = S_{n'm'k'l'} \neq 0$.

Similarly to the fluctuations of $F_2^A(t)$, we assume that the nonzero $S_{nmkl}$'s are unique up to trivial permutations $n \longleftrightarrow k$ and $m \longleftrightarrow l$ that leave $S_{nmkl}$ invariant. In other words, for nonzero $S_{nmkl}$, only the following $S_{nmkl}$ are equal,

$$S_{nmkl} = S_{kmnl} = S_{klnm} = S_{nlkm}.$$

Using this assumption, we can reduce the constraint into four sums,

$$\Delta_{F_A^4}^2 = \sum\nolimits'_{n,m,k,l} T_{nmkl} T_{nmkl}^* + \sum\nolimits'_{n,m,k,l} T_{nmkl} T_{kmnl}^* \\ + \sum\nolimits'_{n,m,k,l} T_{nmkl} T_{klnm}^* + \sum\nolimits'_{n,m,k,l} T_{nmkl} T_{nlkm}^*, \qquad (18)$$

where the prime over the sum includes all constraints on $(nmkl)$ that ensure no double counting between the permutations.

To obtain the bound for $\Delta_{F_4^A}^2$, we show (see Appendix A) that the first term on the right-hand-side of Eq. (18) is dominating, and write it in the form

$$\sum_{n,m,k,l} T_{nmkl} T_{nmkl}^* = \text{tr}(\omega \hat{A} \, \omega_{AAA} \hat{A}^\dagger), \qquad (19)$$

where $\omega$ is defined in Eq. (11), and

$$\omega_{AAA} = \sum_{m,k,l} \hat{P}_m \hat{A} \hat{P}_k \hat{A} \hat{P}_l \hat{A} \left| \Psi_0 \right\rangle \left\langle \Psi_0 \right| \hat{A}^\dagger \hat{P}_l \hat{A}^\dagger \hat{P}_k \hat{A}^\dagger \hat{P}_m. \qquad (20)$$

As shown in Appendix A, we can bound the fluctuations as

$$\Delta_{F_4^A}^2 \leq 4 \left\| \hat{A} \right\|^8 \sqrt{\text{tr}(\omega^2)}. \qquad (21)$$

This is similar to the result in Eq. (13) and implies again that if $\text{IPR}_0$ is proportional to the dimension of the Hilbert space of a many-body quantum system, then the fluctuations decay exponentially with the system size.

## III. MODELS AND INITIAL STATES

In this section, we describe the models and initial states that we use to numerically confirm in Sec. IV that the bounds derived above are tight. We consider three spin-1/2 chains as described next.

(i) The XX model,

$$\hat{H}_{\text{XX}} = \sum_{i=1}^{L-1} J \left( \hat{S}_i^x \hat{S}_{i+1}^x + \hat{S}_i^y \hat{S}_{i+1}^y \right) + h_b \hat{S}_1^z, \qquad (22)$$

where $\hat{S}_i^{x,y,z} = \sigma_i^{x,y,z}/2$ are spin-1/2 operators operating on site $i$ and $\sigma_i^{x,y,z}$ are Pauli matrices, $L$ is the size of the system, $J$ is the coupling strength, and $h_b$ is a border defect, which we introduce to break parity and spin-reversal symmetries [83], without breaking the integrability of the model. Throughout the work, we set $J = 1$ and $h_b = 0.1$. This model can be exactly mapped to noninteracting fermions using the Jordan-Wigner transformation.

(ii) The XXZ model,

$$\hat{H}_{\text{XXZ}} = \hat{H}_{\text{XX}} + J_z \sum_{i=1}^{L-1} \hat{S}_i^z \hat{S}_{i+1}^z, \qquad (23)$$

which is integrable via the Bethe ansatz [84]. We set the anisotropy parameter to $J_z = 0.48$ to avoid additional symmetries that appear for $J = J_z$ and at the roots of unit, such as $J_z = J/2$ [85].

(iii) The XXZ model in Eq. (23) with additional next-nearest neighbor (NNN) couplings,

$$\hat{H}_{\text{NNN}} = \hat{H}_{\text{XXZ}} \\ + \lambda \sum_{i=1}^{L-2} \left[ J \left( \hat{S}_i^x \hat{S}_{i+2}^x + \hat{S}_i^y \hat{S}_{i+2}^y \right) + J_z \hat{S}_i^z \hat{S}_{i+2}^z \right]. \qquad (24)$$

We choose $\lambda = 1$, which guarantees that the system is chaotic.

We use open-boundary conditions for the three models to break translational symmetry. All of them conserve the total magnetization in the $z$-direction, $\hat{S}_{\text{tot}}^z = \sum_i \hat{S}_i^z$. We work within the $\hat{S}_{\text{tot}}^z = 0$ subspace, which has the Hilbert space dimension $D = \binom{L}{L/2}$.

Interacting Hamiltonians without symmetries, like those defined in Eqs. (23) and (24), typically satisfy the condition of non-degenerate energy gaps, which is needed for the bounds in Eq. (13) and Eq. (21). This contrasts with noninteracting models, or systems that can be mapped to free fermions, such as the XX model in Eq. (22), which have a large number of gap degeneracies even when the spectrum is non-degenerate.

### A. Initial States

In this work, we use three initial states: a random state drawn from the Haar measure, which we refer to as the

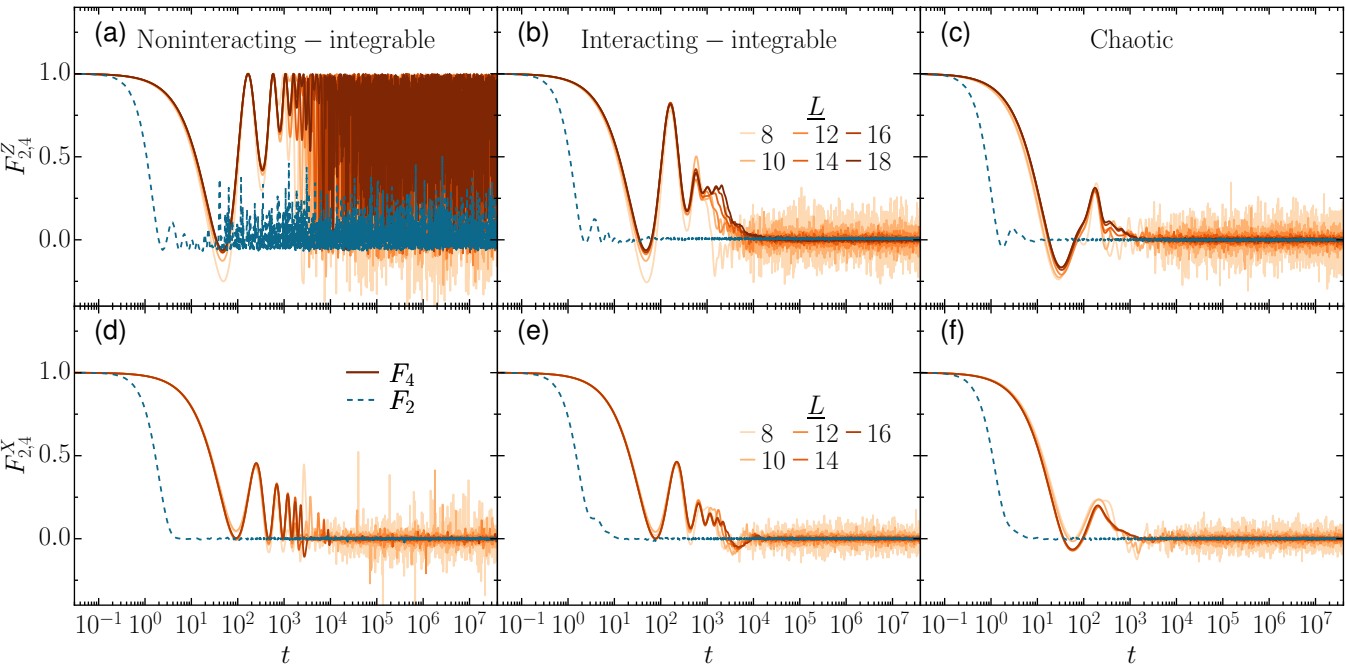

FIG. 1. Dynamics of the correlators $F_{2,4}^Z(t)$ in the top row (a)-(c) and $F_{2,4}^X(t)$ in bottom row (d)-(f) evaluated numerically for the Haar state initial condition and three spin-1/2 models. Solid lines correspond to $F_4^{X,Z}(t)$ and dashed lines to $F_2^{X,Z}(t)$. For $F_4^Z(t)$, a number of system sizes are considered, where the larger sizes are indicated with darker colors [legend in (b) is for (a)-(c) and legend in (e) for (d)-(f)]. For $F_2^Z(t)$, the system size is $L = 18$ and for $F_2^X(t)$, it is $L = 16$. The XX model from Eq. (22) is depicted in the left column [(a), (d)], the XXZ model from Eq. (23) in the middle column [(b), (e)], and the chaotic NNN model from Eq. (24) in the right column [(c), (f)].

Haar state, the Néel state $|\Psi_0\rangle = |\uparrow\downarrow\uparrow\downarrow \ldots \uparrow\downarrow\rangle$, and the domain-wall state $|\Psi_0\rangle = |\uparrow\uparrow\uparrow \ldots \downarrow\downarrow\downarrow\rangle$, where half of the chain has the spins pointing up in the $z$-direction and the other half is pointing down. The Haar state corresponds to an infinite-temperature state with energy in the middle of the many-body spectrum. The Néel and domain-wall states are pure states widely used in experiments, whose energies depend on the model considered (see Appendix B).

## IV. NUMERICAL RESULTS

In this section, we show that when the bound that we derived in Sec. II applies, it is tight. We numerically investigate the evolution and temporal fluctuations of the time-ordered and out-of-time-ordered correlation functions for

$$\hat{A} = \sigma_{L/2}^x \qquad \text{and} \qquad \hat{A} = \sigma_{L/2}^z. \qquad (25)$$

The corresponding correlators are denoted by $F_{2,4}^X$ and $F_{2,4}^Z$, respectively.

We consider the three models presented above and take a random state from the Haar measure as our initial state. The results do not change qualitatively for the other initial states listed in Sec. III A, but they differ in details (see Appendix B).

Since we are interested in the long-time dynamics, we use exact diagonalization, which limits the accessible system's size to $L \leq 18$.

## A. Results

Before analyzing the temporal fluctuations, we present in Fig. 1, the evolution of $F_{2,4}^Z(t)$ [(a)-(c)] and $F_{2,4}^X(t)$ [(d)-(f)] from time $t = 0$ up to their saturation; for the XX, XXZ, and NNN models; for various system sizes. With the exception of $F_{2,4}^Z(t)$ for the XX model [Fig. 1(a)], the correlation functions saturate to a very small value and exhibit small temporal fluctuations that decrease as the system size increases. The fluctuations of $F_{2,4}^X(t)$ are noticeably smaller than those of $F_{2,4}^Z(t)$. This is likely caused by the conservation of the total $z$-magnetization in the studied models, which is also known to slow down the decay in time of $F_4^Z(t)$ [70].

The behavior of $F_4^Z(t)$ after saturation for the XX model [Fig. 1(a)] stands out. While for the other models, $F_4^Z(t)$ fluctuates around zero at long times, for the XX model, it approaches 1, as discussed also in Ref. [26]. In contrast, $F_2^Z(t)$ for the XX model [Fig. 1(a)] does decay to zero, even though it exhibits larger fluctuations than for the XXZ [Fig. 1(b)] and NNN [Fig. 1(c)] models. In Sec. V, we elucidate the behavior of $F_{2,4}^Z(t)$ for the XX

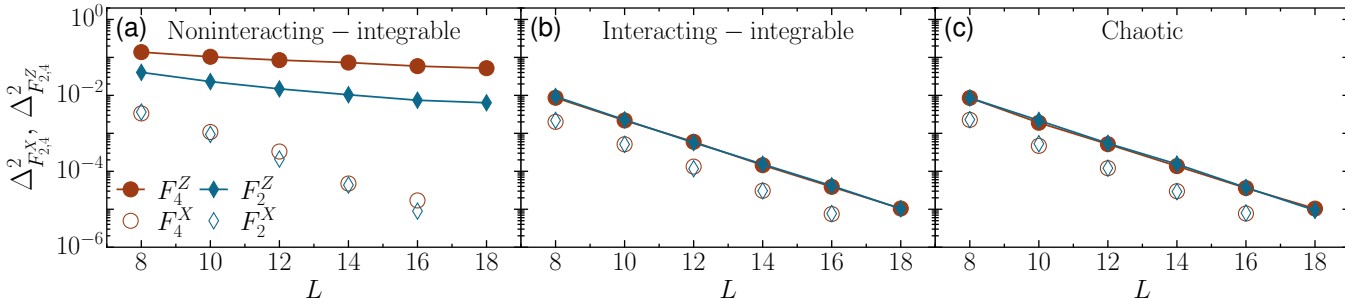

FIG. 2. Variance of the temporal fluctuations of $F_4^{X,Z}(t)$ (circles) and $F_2^{X,Z}(t)$ (diamonds) for the Haar state initial condition as a function of system size, $L$. Filled symbols represent $\Delta^2_{F_{2,4}^Z}$ and empty symbols correspond to $\Delta^2_{F_{2,4}^X}$. The models are (a) the XX defined in Eq. (22), (b) the XXZ from Eq. (23), and (c) the NNN in Eq. (24).

model by analytical arguments.

The saturation of $F_4^X(t)$ for the XX model [Fig. 1(d)] is preceded by quasi-periodic oscillations, which are absent for the chaotic NNN model [Fig. 1(f)]. Smaller oscillations at intermediate times are also visible for the XXZ model, although they happen at a plateau [Fig. 1(e)]. Interestingly, a plateau that gets longer as $L$ increases also appears for $F_4^Z(t)$ in the XXZ model [Fig. 1(b)]. Contrary to $F_4^X(t)$, the time-ordered correlation function $F_2^X(t)$ decays fast to a small value without oscillations for all three models.

With the general picture of the behavior of $F_{2,4}^{X,Z}(t)$ provided in Fig. 1, we study the dependence of the temporal fluctuations, $\Delta_{F_{2,4}^{X,z}}$ as a function of system size. Fig. 2 shows this dependence for the Haar state (for a comparison with the results for the Néel and domain wall states, see Appendix B), and confirms that the bound derived in Sec. II is tight. Namely, that $\Delta_{F_{2,4}^{X,z}}$ decays exponentially with system size for the interacting-integrable [Fig. 2(b)] and chaotic [Fig. 2(c)] systems. For a given chain length, even the magnitude of the fluctuations is comparable for both models.

The behavior for the XX model [Fig. 2(a)] is distinct. While both $\Delta^2_{F_2^X}$ and $\Delta^2_{F_4^X}$ decay exponentially with $L$, $\Delta^2_{F_{2,4}^Z}$ decreases slower than exponentially. As stated in Sec. II, this model has gap degeneracies, so the proof of Sec. II does not apply to it. Nevertheless, since it can be mapped to free fermions, in the next section, we provide numerical and analytical results for the dependence of $\Delta^2_{F_{2,4}^Z}$ on $L$, as well as for the infinite-time averages $\overline{F_{2,4}^Z}$.

## V. XX MODEL

Since the XX model maps exactly to noninteracting fermions, its infinite-time average, $\overline{F_{2,4}^Z}$, and temporal fluctuations, $\Delta^2_{F_{2,4}^Z}$, can be computed analytically. The numerical analysis of the previous section can also be extended to much larger system sizes, because the complexity of the calculations increases only polynomially with

system size.

We compute $F_2^Z(t)$ and $F_4^Z(t)$ by writing them in terms of fermionic creation, $\hat{c}_{L/2}^\dagger$, and annihilation, $\hat{c}_{L/2}$, operators acting on site $L/2$, that is,

$$\sigma_{L/2}^z = 2\hat{c}_{L/2}^\dagger \hat{c}_{L/2} - 1. \tag{26}$$

For quadratic Hamiltonians, the time-evolution of the creation and annihilation operators is given by

$$\hat{c}_i^\dagger(t) = \sum_l u_{il}^*(t)\,\hat{c}_l^\dagger,$$
$$\hat{c}_i(t) = \sum_k u_{ik}(t)\,\hat{c}_k, \tag{27}$$

where $u_{ik}(t) = \langle i|e^{-ih_s t}|k\rangle$ is the single-particle propagator and $h_s$ the single-particle Hamiltonian.

After expressing $F_{2,4}^Z(t)$ in terms of fermionic operators, we calculate the correlations using Wick's theorem (see Appendix C). For generic initial states, the final expression is cumbersome, however, it considerably simplifies at infinite temperature, yielding

$$F_2^Z(t) = \left|u_{L/2,L/2}(t)\right|^2, \tag{28}$$

and

$$F_4^Z(t) = 4\left(\left|u_{L/2,L/2}(t)\right|^4 - \left|u_{L/2,L/2}(t)\right|^2\right) + 1. \tag{29}$$

We can then obtain the infinite time-average,

$$\overline{F_2^Z} = \sum_{\alpha,\beta} \overline{e^{i(\varepsilon_\beta - \varepsilon_\alpha)t}}\left|\langle\tfrac{L}{2}|\alpha\rangle\right|^2\left|\langle\tfrac{L}{2}|\beta\rangle\right|^2 = \sum_\alpha \left|\langle\tfrac{L}{2}|\alpha\rangle\right|^4, \tag{30}$$

where $|\alpha\rangle$ and $\varepsilon_\alpha$ are the eigenstates and eigenvalues of $h_s$. In the last equality, we assumed that the single-particle spectrum is non-degenerate, which is true for the XX model with the border impurity. We see that $\overline{F_2^Z}$ is the IPR of the initial state corresponding to a single particle at the center of the chain, $L/2$, and projected in the

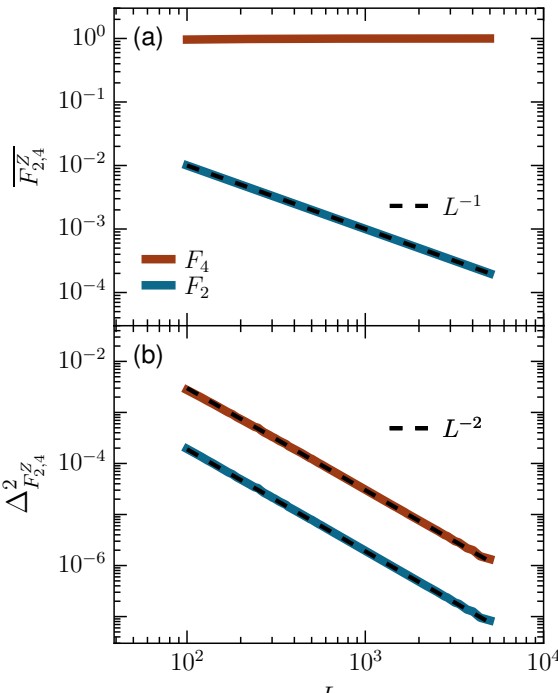

FIG. 3. (a) Infinite-time average $\overline{F_4^Z}$ and (b) temporal fluctuations after saturation, $\Delta_{F_{2,4}^Z}$, as a function of system size; for the XX model and an initial state at infinite temperature. Solid lines represent the numerical data and dashed lines, the analytical results.

basis of the single-particle eigenstates of $h_s$. For delocalized single-particle initial states, we have that $\overline{F_2^Z} \propto L^{-1}$.

Using $\overline{F_2^Z}$, we write the temporal fluctuations of $F_2^Z(t)$ as

$$F_2^Z(t) - \overline{F_2^Z} = \sum_{\alpha \neq \beta} |\langle \alpha|i\rangle|^2 |\langle \beta|i\rangle|^2 e^{i(\varepsilon_\beta - \varepsilon_\alpha)t}, \quad (31)$$

and obtain the variance

$$\Delta_{F_2^Z}^2 = \overline{\left| F_2^Z - \overline{F_2^Z} \right|^2} = \sum_{\alpha \neq \beta} |\langle \alpha|i\rangle|^4 |\langle \beta|i\rangle|^4 \sim \frac{1}{L^2}. \quad (32)$$

A similar derivation follows for the temporal fluctuations of $F_4^Z(t)$. The infinite-time average of the out-of-time ordered correlation function is given by

$$\overline{F_4^Z} = 1 + 4 \sum_{\alpha \neq \beta} \left| \langle \tfrac{L}{2}|\alpha\rangle \right|^4 \left| \langle \tfrac{L}{2}|\beta\rangle \right|^4$$

$$- 4 \sum_\alpha \left| \langle \tfrac{L}{2}|\alpha\rangle \right|^4 \sim 1 - \frac{4}{L} + O\left(L^{-2}\right), \quad (33)$$

a result that was first discussed in Ref. [26]. Since $F_4^Z(t)$ is dominated by the second term in the parenthesis of Eq. (29), its temporal fluctuations decay with system size similarly to what happens for $F_2^Z(t)$ [cf. Eq. (28)].

To confirm these analytical estimates numerically, we compute $F_2^Z(t)$ and $F_4^Z(t)$ using Eq. (28) and Eq. (29) for a number of system sizes. Figure 3(a) shows that the infinite-time average decays as $L^{-1}$, and Fig. 3(b) shows that the temporal fluctuations around that average decay as $L^{-2}$, in perfect alignment with the analytical estimates.

The calculation of the temporal fluctuations of $F_{2,4}^X(t)$ is considerably harder, because both correlators expressed in terms of fermionic operators are non-local, so one has to perform Wick's contraction of the order of $L$ operators. While this can be done numerically, we were not able to obtain analytical estimates, which would help to explain the exponential decay of the fluctuations with the system size.

## VI. DISCUSSION

We rigorously showed, that for any quantum system with non-degenerate energy gaps, the temporal fluctuations around the saturation value of time-ordered and out-of-time-ordered correlation functions are bounded by the square root of the inverse participation ratio of the initial state. Since most physical initial states are composed of exponentially (in the system size $L$) many eigenstates of the Hamiltonian, this implies that for such initial states, the temporal fluctuations decay at least exponentially with $L$. We verified numerically that the bounds are tight; the fluctuations decay exponentially for the interacting integrable XXZ spin-1/2 chain and for its chaotic version with next-nearest-neighbor couplings. Our results demonstrate that the decay of the temporal fluctuations of correlators as a function of the system size cannot serve as a reliable metric of chaoticity. This has to be contrasted with the decay of temporal fluctuations when the initial state is one eigenstate of the Hamiltonian. For such initial states, the fluctuations are related to the off-diagonal matrix elements, which *do* show different decay with system size for chaotic and integrable systems [86].

The only distinguishing behavior that we identified was for the 1D XX model, which is exactly mappable to non-interacting fermions. Since this model has degenerate energy gaps, the bounds that we obtained on the decay of the fluctuations do not apply; however, they can be calculated both analytically and numerically. We find that for this system, the temporal fluctuations of the time-ordered and out-of-time-ordered correlation functions in the $z$-direction, $F_{2,4}^Z(t)$, decay as $L^{-2}$. On the other hand, we observed numerically that the temporal fluctuations of $F_{2,4}^X(t)$ decay exponentially with system size. We argue that, in this case, the exponential decay stems from the non-locality of $F_{2,4}^X(t)$, when written in terms of fermionic annihilation and creation operators. It remains to put this claim on more solid ground. It would also be interesting to investigate if there are scenarios in which different power-law decays of the fluctuations are

possible.

ACKNOWLEDGMENTS

This research was supported by a grant from the United States-Israel Binational Foundation (BSF, Grant No. 2019644), Jerusalem, Israel, and the United States National Science Foundation (NSF, Grant No. DMR-1936006). LFS thanks Vinitha Balachandran, Marcos Rigol, and Dario Poletti for valuable discussions.

## Appendix A: Proofs of the general bounds

In this section, we provide detailed proofs of the bounds in Eqs. (13) and (21) of the main text.

We showed in Eq. (10) that $\Delta^2_{F_2^A}$ can be bound by

$$\Delta^2_{F_2^A} \leq \text{tr}\Big(\hat{A}\omega_A\hat{A}^\dagger\omega\Big), \tag{A1}$$

where $\omega$ and $\omega_A$ are defined in Eq. (11). Using the Cauchy-Schwarz inequality,

$$\text{tr}\left(V^\dagger W\right) \leq \sqrt{\text{tr}\left(V^\dagger V\right)}\sqrt{\text{tr}\left(W^\dagger W\right)}, \tag{A2}$$

setting $V = \hat{A}\omega_A$ and $W = \hat{A}^\dagger\omega$, and using the cyclic property of the trace, we further bound Eq. (10) as follows,

$$\begin{aligned}
\Delta^2_{F_2^A} &\leq \sqrt{\text{tr}\left(\hat{A}^\dagger\hat{A}\omega_A^2\right)}\sqrt{\text{tr}\left(\hat{A}\hat{A}^\dagger\omega^2\right)} \\
&\leq \|\hat{A}\|^2\sqrt{\text{tr}\left(\omega_A^2\right)}\sqrt{\text{tr}\left(\omega^2\right)},
\end{aligned} \tag{A3}$$

where we used that for two positive matrices $A$ and $B$, $\text{tr}(AB) \leq \|A\|\text{tr}(B)$. We can bound $\text{tr}(\omega_A^2)$ by

$$\begin{aligned}
\text{tr}(\omega_A^2) &= \sum_n \text{tr}\left(\hat{P}_n\hat{A}\left|\Psi_0\right\rangle\left\langle\Psi_0\right|\hat{A}^\dagger\hat{P}_n\hat{A}\left|\Psi_0\right\rangle\left\langle\Psi_0\right|\hat{A}^\dagger\hat{P}_n\right) \\
&\leq \left\|\hat{A}\left|\Psi_0\right\rangle\left\langle\Psi_0\right|\hat{A}^\dagger\right\|\sum_n \text{tr}\left(\hat{P}_n\hat{A}\left|\Psi_0\right\rangle\left\langle\Psi_0\right|\hat{A}^\dagger\right) \\
&\leq \left\|\hat{A}\left|\Psi_0\right\rangle\left\langle\Psi_0\right|\hat{A}^\dagger\right\|\left\langle\Psi_0\right|\hat{A}^\dagger\hat{A}\left|\Psi_0\right\rangle \leq \left\|\hat{A}\right\|^4.
\end{aligned} \tag{A4}$$

This gives

$$\Delta^2_{F_2^A} \leq \left\|\hat{A}\right\|^4\sqrt{\text{tr}(\omega^2)}. \tag{A5}$$

For the fluctuations of $F_4^A$, we start with Eq. (18). We designate a given permutation of $(nmkl)$ by $\sigma(nmkl)$, and notice that it is bijective. Then, using the Cauchy-Schwarz inequality, we have

$$\left|\sum\nolimits'_{nmkl} T_{nmkl}T^*_{\sigma(nmkl)}\right| \leq \sqrt{\sum\nolimits'_{nmkl} T_{nmkl}T^*_{nmkl}\sum\nolimits'_{nmkl} T_{\sigma(nmkl)}T^*_{\sigma(nmkl)}} \leq \sum_{nmkl} T_{nmkl}T^*_{nmkl}, \tag{A6}$$

where in the last inequality we renamed the indexes of the second term in the square root. We also removed the constraints on the sum, using the fact that all elements of the sum are now positive. This allows us to write

$$\Delta^2_{F_4^A} \leq 4\sum_{n,m,k,l} T_{nmkl}T^*_{nmkl}.$$

Using the definitions of $\omega$ in Eq. (11) and $\omega_{AAA}$ in Eq. (20), which are positive operators, we have

$$\sum_{n,m,k,l} T_{nmkl}T^*_{nmkl} = \text{tr}\omega\hat{A}\,\omega_{AAA}\hat{A}^\dagger.$$

Using the Cauchy–Schwarz inequality combined with the cyclic property of the trace,

$$\mathrm{tr}(\omega \hat{A} \omega_{AAA} \hat{A}^\dagger) \leq \sqrt{\mathrm{tr}\left(\omega^2 \hat{A}\hat{A}^\dagger\right) \mathrm{tr}\left(\omega_{AAA}^2 \hat{A}^\dagger \hat{A}\right)} \leq \left\|\hat{A}\right\|^2 \sqrt{\mathrm{tr}(\omega^2)\mathrm{tr}(\omega_{AAA}^2)}. \tag{A7}$$

We now proceed by bounding $\mathrm{tr}(\omega_{AAA}^2)$, which is given by

$$\mathrm{tr}(\omega_{AAA}^2) = \sum_{m,k,l}\sum_{k',l'}\mathrm{tr}\left[\left(\hat{P}_{l'}\hat{A}^\dagger\hat{P}_{k'}\hat{A}^\dagger\hat{P}_m\hat{A}\hat{P}_k\hat{A}\hat{P}_l\hat{A}\left|\Psi_0\right\rangle\left\langle\Psi_0\right|\hat{A}^\dagger\hat{P}_l\hat{A}^\dagger\hat{P}_k\hat{A}^\dagger\hat{P}_m\hat{A}\hat{P}_{k'}\hat{A}\hat{P}_{l'}\right)\hat{A}\left|\Psi_0\right\rangle\left\langle\Psi_0\right|\hat{A}^\dagger\right]. \tag{A8}$$

Since $\hat{A}\left|\Psi_0\right\rangle\left\langle\Psi_0\right|\hat{A}^\dagger$ and the matrix to the left of it are positive, we can write

$$\mathrm{tr}(\omega_{AAA}^2) \leq \left\|\hat{A}\right\|^2 \sum_{m,k,l}\sum_{k',l'}\mathrm{tr}\left[\hat{P}_{l'}\hat{A}^\dagger\hat{P}_{k'}\hat{A}^\dagger\hat{P}_m\hat{A}\hat{P}_k\hat{A}\hat{P}_l\hat{A}\left|\Psi_0\right\rangle\left\langle\Psi_0\right|\hat{A}^\dagger\hat{P}_l\hat{A}^\dagger\hat{P}_k\hat{A}^\dagger\hat{P}_m\hat{A}\hat{P}_{k'}\hat{A}\hat{P}_{l'}\right], \tag{A9}$$

and eliminate the sum over $l'$ using the cyclic property of the trace and the fact that $\sum_m P_m = I$. Therefore,

$$\mathrm{tr}(\omega_{AAA}^2) \leq \left\|\hat{A}\right\|^2 \sum_{m,k,l}\sum_{k'}\mathrm{tr}\left[\hat{P}_{k'}\hat{A}^\dagger\hat{P}_m\hat{A}\hat{P}_k\hat{A}\hat{P}_l\hat{A}\left|\Psi_0\right\rangle\left\langle\Psi_0\right|\hat{A}^\dagger\hat{P}_l\hat{A}^\dagger\hat{P}_k\hat{A}^\dagger\hat{P}_m\hat{A}\hat{P}_{k'}\hat{A}\hat{A}^\dagger\right]. \tag{A10}$$

We can now proceed along the same lines as before until we get

$$\mathrm{tr}(\omega_{AAA}^2) \leq \left\|\hat{A}\right\|^{12}. \tag{A11}$$

Combining the expressions, we obtain

$$\Delta_{F_4^A}^2 \leq 4\left\|\hat{A}\right\|^8 \sqrt{\mathrm{tr}(\omega^2)}. \tag{A12}$$

## Appendix B: Dependence on the initial state

Here, we compare the results for the variance of the temporal fluctuations obtained for the Haar state as the initial state with those for the Néel and the domain-wall states. The expectation of the Hamiltonian, $E_0 = \left\langle\Psi_0\right|H\left|\Psi_0\right\rangle$ of these two states depends on the model according to [87],

$$\text{Néel state:} \quad E_0 \ = \ \frac{J_z}{4}\left[-(L-1) + \lambda(L-2))\right],$$

$$\text{Domain wall state:} \quad E_0 \ = \ \frac{J_z}{4}\left[(L-3) + \lambda(L-6))\right].$$

For the XX model, where $J_z, \lambda = 0$, the energies of all three states are in the middle of the spectrum, $E_0 = 0$. For the XXZ model with large $L$ and for our choices of parameters, $|E_0| \sim JL/8$, where $E_0$ is negative (positive) for the Néel (domain wall) state. For the NNN model, the energy of the Néel state is close to the middle of the spectrum, $E_0 \sim 0$, where chaos is strong and the eigenstates are very delocalized, while the energy of the domain wall state for large $L$ is closer to the positive edge of the spectrum, $E_0 \sim JL/4$.

In Fig. 4, we plot $\Delta_{F_{2,4}^{X,Z}}^2$ for the three different initial states. Their results are qualitatively similar, but there are subtle differences associated with the position of the energy of the initial state in the spectrum. While for the Haar state [Fig. 4(a),(d)], there is practically no difference in the values of $\Delta_{F_4^{X,Z}}^2$ for the interacting-integrable and the chaotic model, the fluctuations for the Néel state [Fig. 4(b),(e)] are smaller for the chaotic model, since in this case, $|\Psi_0\rangle$ is in the middle of the spectrum, being thus more delocalized than for the XXZ model. A similar explanation can be given for the smaller fluctuations associated with the Néel state for the NNN model [Fig. 4(b),(e)] when compared with those for the domain wall state under the same model [Fig. 4(c),(f)].

With respect to the XX model, the magnitude of the fluctuations is analogous for all three initial states. The difference lies in the operator considered, as already discussed in Fig. 2(a), that is, $\Delta_{F_4^Z}^2$ decreases slower than exponentially [Fig. 4(a)-(c)] and $\Delta_{F_4^X}^2$ decays exponentially with $L$ [Fig. 4(d)-(f)].

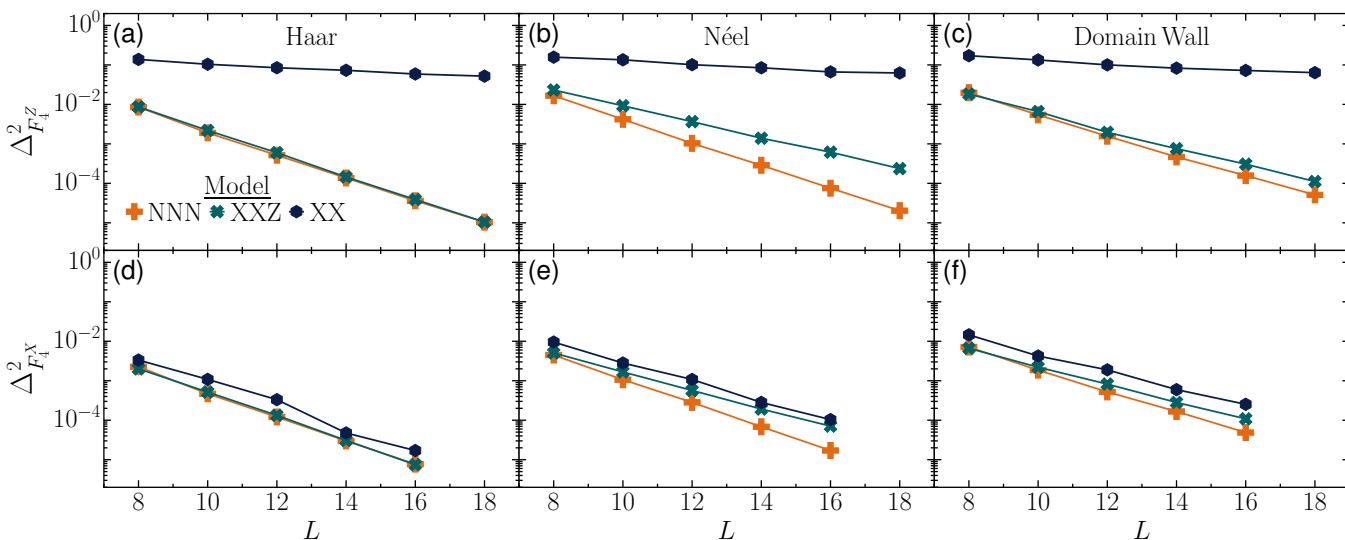

FIG. 4. System-size scaling of the variance of the temporal fluctuations of $F_4^Z(t)$ $[F_4^X(t)]$ for the (a) [(d)] Haar, (b) [(e)] Néel, and (c) [(f)] domain-wall states for the XX [Eq. (22)], XXZ [Eq. (23)], and NNN [Eq. (24)] models.

## Appendix C: Detailed derivations for the XX model

### 1. Out-of-time-order correlation function

In the following, we present the detailed derivation of $F_4^Z(t)$ defined in Eq. (29) of the main text,

$$F_4^Z(t) = \left\langle \hat{W}(t)\hat{V}(0)\hat{W}(t)\hat{V}(0) \right\rangle, \tag{C1}$$

where $\hat{W}(t) = \sigma_i^z(t)$ and $\hat{V} = \sigma_j^z$. We rewrite $\hat{W}$ and $\hat{V}$ in terms of fermionic operators as

$$\sigma_i^z(t) = 2\hat{n}_i(t) - 1 \quad \text{and} \quad \sigma_j^z = 2\hat{n}_j - 1. \tag{C2}$$

where $\hat{n}_i(t) = \hat{c}_i^\dagger(t)\hat{c}_i(t)$ and $\hat{n}_j = \hat{c}_j^\dagger \hat{c}_j$, so that

$$\begin{aligned} F_4^Z(t) &= \langle (2\hat{n}_i(t) - 1)(2\hat{n}_j - 1)(2\hat{n}_i(t) - 1)(2\hat{n}_j - 1) \rangle \\ &= \langle (4\hat{n}_i(t)\hat{n}_j - 2\hat{n}_i(t) - 2\hat{n}_j + 1)(4\hat{n}_i(t)\hat{n}_j - 2\hat{n}_i(t) - 2\hat{n}_j + 1) \rangle. \end{aligned} \tag{C3}$$

Expanding the previous expression, some terms trivially cancel and it gets reduced to

$$F_4^Z(t) = \langle 16\hat{n}_i(t)\hat{n}_j\hat{n}_i(t)\hat{n}_j - 8\hat{n}_i(t)\hat{n}_j\hat{n}_i(t) - 8\hat{n}_j\hat{n}_i(t)\hat{n}_j - 4\hat{n}_i(t)\hat{n}_j + 4\hat{n}_j\hat{n}_i(t) \rangle + 1, \tag{C4}$$

where we used $\hat{n}_i^2(t) = \hat{n}_i(t)$ and $\hat{n}_j^2 = \hat{n}_j$.

In the following, we expand the expectation value and work on each of the terms above. But before doing that, let us express the time evolution of $\hat{c}_i^\dagger(t)$ and $\hat{c}_i(t)$ as

$$\begin{aligned} \hat{c}_i^\dagger(t) &= \sum_l u_{il}^*(t)\hat{c}_l^\dagger \\ \hat{c}_i(t) &= \sum_k u_{ik}(t)\hat{c}_k, \end{aligned} \tag{C5}$$

where $u_{ik}(t) = \langle i | e^{-ih_s t} | k \rangle$ is the single-particle propagator and $h_s$ the single-particle Hamiltonian.

Using Eq.(C5), we then express the first term of Eq.(C4) as

$$\langle \hat{n}_i(t)\hat{n}_j\hat{n}_i(t)\hat{n}_j \rangle = \sum_{klpq} u_{ik}(t)u_{il}^*(t)u_{ip}(t)u_{iq}^*(t) \left\langle \hat{c}_l^\dagger \hat{c}_k \hat{c}_j^\dagger \hat{c}_j \hat{c}_q^\dagger \hat{c}_p \hat{c}_j^\dagger \hat{c}_j \right\rangle \tag{C6}$$

Using Wick's theorem, we expand the expectation value in terms of nonzero pairwise contractions. After that, reintroducing the time dependence gives,

$$
\begin{aligned}
\langle \hat{n}_i(t)\,\hat{n}_j\hat{n}_i(t)\,\hat{n}_j \rangle \;=\; & \langle \hat{n}_i(t) \rangle \quad \Big\{ \langle \hat{n}_j \rangle \Big[ \langle \hat{n}_i(t) \rangle \langle \hat{n}_j \rangle + \big\langle \hat{c}_i^\dagger(t)\,\hat{c}_j \big\rangle \big\langle \hat{c}_i(t)\,\hat{c}_j^\dagger \big\rangle \Big] \\
& + \big\langle \hat{c}_j^\dagger \hat{c}_i(t) \big\rangle \Big[ \big\langle \hat{c}_j \hat{c}_i^\dagger(t) \big\rangle \langle \hat{n}_j \rangle - \big\langle \hat{c}_i^\dagger(t)\,\hat{c}_j \big\rangle + \langle \hat{n}_j \rangle \big\langle \hat{c}_i^\dagger(t)\,\hat{c}_j \big\rangle \Big] \\
& + \langle \hat{n}_j \rangle \Big[ \big\langle \hat{c}_j \hat{c}_i^\dagger(t) \big\rangle \big\langle \hat{c}_i(t)\,\hat{c}_j^\dagger \big\rangle + \langle \hat{n}_i(t) \rangle - \langle \hat{n}_j \rangle \langle \hat{n}_i(t) \rangle \Big] \Big\} \\
+\; & \big\langle \hat{c}_i^\dagger(t)\,\hat{c}_j \big\rangle \Big\{ \big\langle \hat{c}_i(t)\,\hat{c}_j^\dagger \big\rangle \Big[ \langle \hat{n}_i(t) \rangle \langle \hat{n}_j \rangle + \big\langle \hat{c}_i^\dagger(t)\,\hat{c}_j \big\rangle \big\langle \hat{c}_i(t)\,\hat{c}_j^\dagger \big\rangle \Big] \\
& - (1 - \langle \hat{n}_i(t) \rangle) \Big[ \big\langle \hat{c}_j^\dagger \hat{c}_i(t) \big\rangle \langle \hat{n}_j \rangle + \langle \hat{n}_j \rangle \big\langle \hat{c}_i(t)\,\hat{c}_j^\dagger \big\rangle \Big] \\
& - \big\langle \hat{c}_i(t)\,\hat{c}_j^\dagger \big\rangle \Big[ - \big\langle \hat{c}_j^\dagger \hat{c}_i(t) \big\rangle \big\langle \hat{c}_i^\dagger(t)\,\hat{c}_j \big\rangle + \langle \hat{n}_j \rangle \langle \hat{n}_i(t) \rangle \Big] \Big\} \\
+\; & \langle \hat{n}_i(t) \rangle \quad \Big\{ \big\langle \hat{c}_i(t)\,\hat{c}_j^\dagger \big\rangle \Big[ \big\langle \hat{c}_j \hat{c}_i^\dagger(t) \big\rangle \langle \hat{n}_j \rangle - \big\langle \hat{c}_i^\dagger(t)\,\hat{c}_j \big\rangle + \langle \hat{n}_j \rangle \big\langle \hat{c}_i^\dagger(t)\,\hat{c}_j \big\rangle \Big] \\
& + (1 - \langle \hat{n}_i(t) \rangle) \Big[ \langle \hat{n}_j \rangle^2 + \langle \hat{n}_j \rangle - \langle \hat{n}_j \rangle^2 \Big] \\
& - \big\langle \hat{c}_i(t)\,\hat{c}_j^\dagger \big\rangle \Big[ \langle \hat{n}_j \rangle \big\langle \hat{c}_i^\dagger(t)\,\hat{c}_j \big\rangle + \langle \hat{n}_j \rangle \big\langle \hat{c}_j \hat{c}_i^\dagger(t) \big\rangle \Big] \Big\} \\
+\; & \big\langle \hat{c}_i^\dagger(t)\,\hat{c}_j \big\rangle \Big\{ \big\langle \hat{c}_i(t)\,\hat{c}_j^\dagger \big\rangle \Big[ \big\langle \hat{c}_j \hat{c}_i^\dagger(t) \big\rangle \big\langle \hat{c}_i(t)\,\hat{c}_j^\dagger \big\rangle + \langle \hat{n}_i(t) \rangle - \langle \hat{n}_j \rangle \langle \hat{n}_i(t) \rangle \Big] \\
& (1 - \langle \hat{n}_i(t) \rangle) \Big[ \langle \hat{n}_j \rangle \big\langle \hat{c}_i(t)\,\hat{c}_j^\dagger \big\rangle - \big\langle \hat{c}_j^\dagger \hat{c}_i(t) \big\rangle + \big\langle \hat{c}_j^\dagger \hat{c}_i(t) \big\rangle \langle \hat{n}_j \rangle \Big] \\
& + \big\langle \hat{c}_i(t)\,\hat{c}_j^\dagger \big\rangle \Big[ \langle \hat{n}_j \rangle \langle \hat{n}_i(t) \rangle + \big\langle \hat{c}_j^\dagger \hat{c}_i(t) \big\rangle \big\langle \hat{c}_j \hat{c}_i^\dagger(t) \big\rangle \Big] \Big\}
\end{aligned}
\tag{C7}
$$

Analogously, for the second and third terms in Eq.(C4),

$$
\begin{aligned}
\langle \hat{n}_i(t)\,\hat{n}_j\hat{n}_i(t) \rangle = & \langle \hat{n}_i(t) \rangle^2 \langle \hat{n}_j \rangle + \langle \hat{n}_i(t) \rangle \big\langle \hat{c}_j^\dagger \hat{c}_i(t) \big\rangle \big\langle \hat{c}_j \hat{c}_i^\dagger(t) \big\rangle \\
& + \langle \hat{n}_i(t) \rangle \big\langle \hat{c}_i^\dagger(t)\,\hat{c}_j \big\rangle \big\langle \hat{c}_i(t)\,\hat{c}_j^\dagger \big\rangle - \big\langle \hat{c}_i^\dagger(t)\,\hat{c}_j \big\rangle \big\langle \hat{c}_j^\dagger \hat{c}_i(t) \big\rangle + \langle \hat{n}_i(t) \rangle \big\langle \hat{c}_i^\dagger(t)\,\hat{c}_j \big\rangle \big\langle \hat{c}_j^\dagger \hat{c}_i(t) \big\rangle \\
& + \langle \hat{n}_i(t) \rangle \big\langle \hat{c}_i(t)\,\hat{c}_j^\dagger \big\rangle \big\langle \hat{c}_j \hat{c}_i^\dagger(t) \big\rangle + \langle \hat{n}_i(t) \rangle \langle \hat{n}_j \rangle - \langle \hat{n}_i(t) \rangle^2 \langle \hat{n}_j \rangle,
\end{aligned}
\tag{C8}
$$

and,

$$
\begin{aligned}
\langle \hat{n}_j\hat{n}_i(t)\,\hat{n}_j \rangle = & \langle \hat{n}_j \rangle^2 \langle \hat{n}_i(t) \rangle + \langle \hat{n}_j \rangle \big\langle \hat{c}_i^\dagger(t)\,\hat{c}_j \big\rangle \big\langle \hat{c}_i(t)\,\hat{c}_j^\dagger \big\rangle \\
& + \langle \hat{n}_j \rangle \big\langle \hat{c}_j^\dagger \hat{c}_i(t) \big\rangle \big\langle \hat{c}_j \hat{c}_i^\dagger(t) \big\rangle - \big\langle \hat{c}_j^\dagger \hat{c}_i(t) \big\rangle \big\langle \hat{c}_i^\dagger(t)\,\hat{c}_j \big\rangle + \langle \hat{n}_j \rangle \big\langle \hat{c}_j^\dagger \hat{c}_i(t) \big\rangle \big\langle \hat{c}_i^\dagger(t)\,\hat{c}_j \big\rangle \\
& + \langle \hat{n}_j \rangle \big\langle \hat{c}_j \hat{c}_i^\dagger(t) \big\rangle \big\langle \hat{c}_i(t)\,\hat{c}_j^\dagger \big\rangle + \langle \hat{n}_j \rangle \langle \hat{n}_i(t) \rangle - \langle \hat{n}_j \rangle^2 \langle \hat{n}_i(t) \rangle.
\end{aligned}
\tag{C9}
$$

For the fourth and fifth terms of Eq.(C4), we have

$$
\langle \hat{n}_i(t)\,\hat{n}_j \rangle = \langle \hat{n}_i(t) \rangle \langle \hat{n}_j \rangle + \big\langle \hat{c}_i^\dagger(t)\,\hat{c}_j \big\rangle \big\langle \hat{c}_i(t)\,\hat{c}_j^\dagger \big\rangle,
\tag{C10}
$$

$$
\langle \hat{n}_j\hat{n}_i(t) \rangle = \langle \hat{n}_j \rangle \langle \hat{n}_i(t) \rangle + \big\langle \hat{c}_j^\dagger \hat{c}_i(t) \big\rangle \big\langle \hat{c}_j \hat{c}_i^\dagger(t) \big\rangle.
\tag{C11}
$$

It is convenient to rewrite all the expressions above using the Green's function

$$
G_{ij}(t) = \big\langle \hat{c}_i^\dagger(t)\,\hat{c}_j \big\rangle.
\tag{C12}
$$

While we can obtain a general equation for any initial state for which the Wick's theorem holds, it is rather cumbersome. In particular, for the infinite-temperature state considered in this work, it considerably simplifies, since $\langle \hat{n}_j \rangle = \langle \hat{n}_i(t) \rangle = 1/2$, and the remaining terms can be readily written in terms of Green's functions as

$$
\big\langle \hat{c}_i(t)\,\hat{c}_j^\dagger \big\rangle = \big\langle \hat{c}_j \hat{c}_i^\dagger(t) \big\rangle^* = G_{ij}^*(t),
\tag{C13}
$$

where we used the cyclic property of the trace and the fact that the density matrix acquires the simple form $\rho = 1/D$ with Hilbert space dimension $D$.

Using this simplifying property together together with Eqs.(C12)-(C13) in Eqs. (C7), (C8)-(C11), and in turn, substituting everything back in Eq.(C4), gives

$$F_4^Z(t) = 16\left(4\left|G_{ij}(t)\right|^4 - \left|G_{ij}(t)\right|^2\right) + 1. \tag{C14}$$

The Green's function at infinite temperature is given by

$$G_{ij}(t) = \left\langle \hat{c}_i^\dagger(t)\hat{c}_j \right\rangle = \sum_l u_{il}^*(t)\left\langle \hat{c}_l^\dagger \hat{c}_j \right\rangle = \sum_l u_{il}^*(t)\frac{1}{2}\delta_{lj} = \frac{1}{2}u_{ij}^*(t), \tag{C15}$$

where we used Eq. (C5). Finally, substituting Eq. (C15) in Eq. (C14), we obtain the expression for $F_4^Z(t)$ at infinite temperature, which we used in the main text,

$$F_4^Z(t) = 4\left(\left|u_{ij}(t)\right|^4 - \left|u_{ij}(t)\right|^2\right) + 1, \tag{C16}$$

where $u_{ij}(t)$ is the single-particle propagator. In the text, we set $i = j = L/2$.

## 2. Time-ordered correlation function

From the previous subsection, one can check that in terms of fermionic operators, the time-ordered correlation function $F_2^Z(t)$ defined in Eq. (28) of the main text is given by

$$F_2^Z(t) = 4\left\langle \hat{n}_i(t)\hat{n}_j \right\rangle - 2\left\langle \hat{n}_i(t) \right\rangle - 2\left\langle \hat{n}_j \right\rangle + 1. \tag{C17}$$

Applying Wick's theorem, we get

$$F_2^Z(t) = 4\left\langle \hat{n}_i(t) \right\rangle\left\langle \hat{n}_j \right\rangle + 4\left\langle \hat{c}_i^\dagger(t)\hat{c}_j \right\rangle\left\langle \hat{c}_i(t)\hat{c}_j^\dagger \right\rangle - 2\left\langle \hat{n}_i(t) \right\rangle - 2\left\langle \hat{n}_j \right\rangle + 1, \tag{C18}$$

which at infinite temperature is simply

$$F_2^Z(t) = 4\left|G_{ij}(t)\right|^2 = \left|u_{ij}(t)\right|^2. \tag{C19}$$

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
