# Peer review of "Temporal fluctuations of correlators in integrable and chaotic quantum systems"

_SciPost Physics_

## Round 2 · Referee Report · Anonymous (Referee 1) · 2023-9-11

Strengths

  1. The paper addresses a topic of current interest: is it possible to define numerical tools, functions of the observables of a system, which may extend the concept of chaos from semiclassical systems to fully quanum ones. It invetigates one recently advanced proposal, and presents numerical and analytical proofs that it is not a suitable choice.

Weaknesses

  1. I did not spot remarkable weaknesses in the paper.

Report

This paper addresses the long-time behavior of the out-of-time commutator (OTOC) between two generic operators W and V (even though, only the particular case of identical operators W = V is actually considered here) in quantum systems.
The OTOC was introduced in literature as a quantifier of the spreading of observables. It was conjectured that it might serve as a useful probe to explore the properties of semiclassically chaotic systems, but it was later dismissed as such since it was realized that its short-time behavior did not distinguish between systems chaotic or regular in the semiclassical regime. Quite recently, interest in the OTOC was revived by the proposal made in the reference 36 cited in the manuscript, to consider instead the long-time behaviour. The authors of ref. 36 argued that the size of of the fluctuations around the infinite-time limiting value of the OTOC could be a way to discriminate between chaotic and regular behavior, with the latter being characterized by large fluctuations.
The purpose of this paper is to test the validity of this proposal.
The section II of the paper (plus the long appendix) presents an analytical study. It demonstrates that bounds do exist for fluctuations of OTOC for generic systems, and that these bounds imply the exponential vanishing of the fluctuations with the system's size, regardless of its chaotic or regular character.
These analytical estimates are confirmed, in section IV, by numerical analysis of three spin-1/2 chains model systems: the XX model, the XXZ model, and the XXZ model supplemented with additional next-nearest neighbor interactions.
The authors show that, on the basis of the investigation of the properties of the OTOC, it is not possible to discriminate between chaotic and regular systems. Thus, this study completes earlier investigations, extending the unsuitability of the OTOC as probe of the chaotic properties to the large-time limit.
The results of the paper, as far as I can judge, appear correct; the topic investigated is timely. Finally, the manuscript is written clearly with extensive bibliographic references.
I have not critiques, just one (perhaps naive) question: the authors present the content of the section II as it were novel. As far as I can judge, it seems to replicate the contents and the results of reference 76. For example, compare Eqns (13,14) of the manuscript, and Eq. 9 of ref. 76. Yet, the authors do not make mention of this, reference 76 is just quickly cited at the end of the section.
I am a bit confused. Could the authors spend some words explaining the differences, if any, between their results in section II and those in ref. 76?

Requested changes

  1. No requested changes, except for one optional amendment mentioned in the report

  • validity: good
  • significance: high
  • originality: high
  • clarity: good
  • formatting: good
  • grammar: good

Author:  Talía L. M. Lezama  on 2023-11-17  [id 4130]

(in reply to Report 2 on 2023-09-11)

We thank the Referee for their positive comments and their careful review of our work. We reply to their questions below.

The Referee writes:
"I have not critiques, just one (perhaps naive) question: the authors present the content of the section II as it were novel. As far as I can judge, it seems to replicate the contents and the results of reference 76. For example, compare Eqns (13,14) of the manuscript, and Eq. 9 of ref. 76. Yet, the authors do not make mention of this, reference 76 is just quickly cited at the end of the section. I am a bit confused. Could the authors spend some words explaining the differences, if any, between their results in section II and those in ref. 76?"

Our reply:
The Referee is correct in asserting that our derivation is closely related to that of Short and Farrelly (Ref. [76]). We drew inspiration from their work and used a similar technique to obtain our results. We intended to make this clear by maintaining their notation and starting Section II with: 'To obtain general bounds on the temporal fluctuations of F^A_2,4, we generalize the results of Refs. [74–76] for the fluctuations in an observable ⟨A(t)⟩.' To make it more explicit, we introduced a new sentence after Eq. (14) to ensure a clearer explanation of the connection and differences between our results and theirs. The main difference is that while Ref [76] considered the fluctuations of an observable, <A(t)>, we focus on the autocorrelation functions <A(t)A(0)> and out-of-time ordered correlation functions <A(t)A(0)A(t)A(0)>. The entire derivation is presented in the Appendix and is slightly different and somewhat more intricate, especially for the case of out-of-time ordered correlation functions. The bounds for both <A(t)A(0)> and <A(t)A(0)A(t)A(0)> are somehow softer compared to Ref. [76], as reflected in the presence of the square-root of the inverse participation ratio, and larger power of the norm of the operators.

---

## Round 2 · Referee Report · Anonymous (Referee 2) · 2023-9-18

Strengths

The paper discusses OTOCS as fashionable tool to characterise many-body quantum systems and their various regimes of complexity. The study is quite exhaustive for spin systems in 1D. Analytical bounds for the fluctuations of time-integrated asymptotic correlation functions are given.

Weaknesses

No systems beyond 1D, no bosonic many-body quantum systems, no finite time analysis.

Report

Time-ordered correlation and out-of-time-ordered correlation functions and their dependence on the system size L are exhaustively analysed. Analytical bounds that are found for the asymptotic temporally integrated signals with fluctuation decay at least exponentially with L. Three spin-1/2 models in 1D lattices are considered: a chaotic model with first and second-neighbor couplings, the integrable interacting XXZ model, and the integrable noninteracting XX model. These are important model cases, yet all 1D and spin systems. The basic question is whether these results generalise to more D systems with or without more interactions (more links, or longer-range interactions). What is also not extremely clear is insofar these results say something about the transient behaviour of quantum many-body systems, i.e. at finite times that is more of experimental interest. How can the predictions made here be observed experimentally?

Requested changes

Maybe some guesses can be given on what would be expected for bosonic and more than 1D systems? Some quantum chaos literature concerning many-body bosonic and 2D or 3D systems maybe cited in addition, see e.g. many papers by Kolovsky et al., Phys. Rev. Lett. 130, 080401 (2023), and Phys. Rev. A 93, 043620 (2016) as a 2D example. The 2D case is interesting because of the possibility to have more and more links between lattice size that govern the chaoticity of the system. The implications for finite times could be explicated please, please see my report.

  • validity: top
  • significance: high
  • originality: high
  • clarity: high
  • formatting: excellent
  • grammar: perfect

Author:  Talía L. M. Lezama  on 2023-11-17  [id 4129]

(in reply to Report 1 on 2023-09-18)

We thank the Referee for their positive comments and constructive report. We address the points raised by the Referee below.

The Referee writes:
"Maybe some guesses can be given on what would be expected for bosonic and more than 1D systems? Some quantum chaos literature concerning many-body bosonic and 2D or 3D systems maybe cited in addition, see e.g. many papers by Kolovsky et al., Phys. Rev. Lett. 130, 080401 (2023), and Phys. Rev. A 93, 043620 (2016) as a 2D example. The 2D case is interesting because of the possibility to have more and more links between lattice size that govern the chaoticity of the system."

Our reply:
We thank the Referee for bringing these works to our attention. The rigorous part of our work is general and valid for any Hamiltonian with a bounded local Hilbert space dimension, including higher dimensional systems. Our results are not strictly valid for bosonic systems where the local Hilbert space is unbounded unless the system is in a parameter regime where the bosonic occupation is low. We have added a comment about this to the discussion and added the pertinent suggested reference. Numerically, we can only verify our bounds for a reasonable number of system sizes in one dimension.

The Referee writes"
"What is also not extremely clear is insofar these results say something about the transient behaviour of quantum many-body systems, i.e. at finite times that is more of experimental interest. How can the predictions made here be observed experimentally?"

Our Reply:
The analysis of short times was done in previous works, as discussed in the Introduction. It was found that both integrable and nonintegrable models can present exponential growth of OTOCs due to unstable points. We do not have a rigorous analytical result for intermediate times, but our numerical results indicate that there is no clear qualitative difference between chaotic and integrable systems.

---

## Editorial Decision

resubmitted